# Physics-Informed Neural Networks with Trust-Region Sequential Quadratic Programming

## Abstract

Physics-Informed Neural Networks (PINNs) represent a significant advancement in Scientific Machine Learning (SciML), which integrate physical domain knowledge into an empirical loss function as soft constraints and apply existing machine learning methods to train the model. However, recent research has noted that PINNs may fail to learn relatively complex Partial Differential Equations (PDEs). This paper addresses the failure modes of PINNs by introducing a novel, hard-constrained deep learning method — trust-region Sequential Quadratic Programming (trSQP-PINN). In contrast to directly training the penalized soft-constrained loss as in PINNs, our method performs a linear-quadratic approximation of the hard-constrained loss, while leveraging the soft-constrained loss to adaptively adjust the trust-region radius. We only trust our model approximations and make updates within the trust region, and such an updating manner can overcome the ill-conditioning issue of PINNs. We also address the computational bottleneck of second-order SQP methods by employing quasi-Newton updates for second-order information, and importantly, we introduce a simple pretraining step to further enhance training efficiency of our method. We demonstrate the effectiveness of trSQP-PINN through extensive experiments. Compared to existing hard-constrained methods for PINNs, such as penalty methods and augmented Lagrangian methods, trSQP-PINN significantly improves the accuracy of the learned PDE solutions, achieving up to 1-3 orders of magnitude lower errors. Additionally, our pretraining step is generally effective for other hard-constrained methods, and experiments have shown the robustness of our method against both problem-specific parameters and algorithm tuning parameters.

## 1 Introduction

Partial Differential Equations (PDEs) are essential mathematical tools used to model a variety of physical phenomena, including heat transfer, fluid dynamics, general relativity, and quantum mechanics. These equations are often derived from fundamental principles, such as the conservation of mass and energy, and describe how physical quantities change over space and time. However, deriving analytical solutions of PDEs in real-world applications is usually infeasible. Numerical methods like finite difference methods (Thomas, 1995), finite element methods (Zienkiewicz et al., 2013), and pseudo-spectral methods (Fornberg, 1996) are commonly employed to approximate solutions. These methods, though effective, require finer meshing of the domain and can be computationally intensive. With the advent of machine learning (ML), there has been a growing interest recently in applying ML methods to solve PDEs. Deep neural networks, known for their large capacity and expressivity, are particularly useful in learning nonlinear mappings between inputs (e.g., spatial/temporal coordinates) and outputs (e.g., measurements of physical quantities), thus leading the way in such applications. This emerging field, called Scientific Machine Learning (SciML), combines data-driven approaches (mostly neural network training) with traditional scientific modeling (i.e., PDEs/ODEs) to address the computational challenges of classical numerical methods and enhance solution accuracy (von Rueden et al., 2021; Willard et al., 2020; Huang et al., 2022).

The Physics-Informed Neural Network (PINN) is a recent development that integrates domain-driven principles with data-driven ML methods for learning PDE solutions (Cuomo et al., 2022; Hao et al., 2022; Raissi et al., 2019). The idea behind PINN is simple: it incorporates PDEs into an empirical loss function as soft constraints, and then optimizes the soft-constrained loss towards zero using

classical (unconstrained) ML methods (e.g., Adam and L-BFGS). PINN is generally applied within a supervised-learning framework, utilizing paired labeled datasets to define the loss. Unlike classical PDE solvers that require meshing, PINN is mesh-free, making it well-suited for large-scale problems with irregular domains. PINN has been successfully employed to various SciML applications, including nano-optics materials (Chen et al., 2020), biomedicine (Sahli Costabal et al., 2020), fluid mechanics (Raissi et al., 2020), optimal control (Mowlavi & Nabi, 2023; Barry-Straume et al., 2022), and inverse design (Lu et al., 2021; Wiecha et al., 2021).

However, recent research has observed some negative results of PINN: it can fail to learn relatively complex PDEs despite employing deep neural network architectures with ample expressivity power (Krishnapriyan et al., 2021). This issue is largely attributed to optimization challenges, including ill-conditioning issue and complex loss landscape, stemming from the presence of differential operators in soft penalty terms within the loss function of PINN (Krishnapriyan et al., 2021; Rathore et al., 2024; Basir, 2022). An imbalance between the empirical loss and the soft penalties can skew training, favoring certain constraints, and thus reducing the accuracy and generalizability of PINN. The nonconvex loss landscape also makes PINN difficult for gradient-based methods to find the optimal solution. Additionally, the complexity of neural network models and their sensitivity to variations in PDEs, coupled with the reliance on fine tuning of hyperparameters, can lead to poor convergence and physically infeasible solutions when applying PINN in practice.

To address the failure modes of PINNs, multiple strategies have recently been proposed, the majority of which attempt to alleviate the ill-conditioning issue and/or improve the loss landscape. For example, Krishnapriyan et al. (2021) proposed two resolutions. In one approach, the authors suggested using curriculum regularization, where the penalty term starts from a simple PDE regularization and becomes progressively more complex as neural network gets trained. Another approach is to pose the problem as a sequence-to-sequence learning task (via spatio-temporal decomposition), rather than learning to predict the entire domain at once. Liu et al. (2024) incorporated a preconditioner into the loss function to mitigate the condition number. Subramanian et al. (2022) introduced an adaptive collection scheme that progressively collects more data points in the regions of high errors. Wang et al. (2021; 2022a;b) reformulated the loss function by reweighting or resampling to balance different regularization components in the loss. Rathore et al. (2024) developed a hybrid optimization method by integrating Newton-CG with Adam and L-BFGS to better adapt to PINN's landscape. While the aforementioned literature has demonstrated the effectiveness of their strategies in learning various PDEs under favorable conditions, they still primarily revolve around regularized, soft-constrained loss functions, which may not yield feasible solutions regardless of how we assign weight to the penalty term. There exist simple examples that, for any finite weight, the soft-constrained loss cannot recover the solution to the hard-constrained problem[1].

Motivated by this concern, another series of literature aims to impose PDEs as hard constraints and apply constrained optimization methods to train the model. For example, Nandwani et al. (2019) converted soft-constrained PINN problems into alternating min-max problems and applied gradient descent ascent methods to solve them. Dener et al. (2020) applied (stochastic) augmented Lagrangian methods to approximate the Fokker-Planck-Landau collision operator, using stochastic gradient descent to solve inner unconstrained minimization subproblems. Subsequently, Lu et al. (2021) proposed using penalty methods and augmented Lagrangian methods within the PINN framework for inverse design problems. In contrast to dealing with the PINN loss directly, the above literature reverts the soft-constrained regularizations back to hard constraints (aligning with our original goals of solving PDEs), and primarily applies either penalty methods[2] or augmented Lagrangian methods to solve the resulting hard-constrained problems.

Although existing hard-constrained methods based on penalty methods and augmented Lagrangian methods have shown significant improvements over PINNs, these two methods have largely been supplanted by Sequential Quadratic Programming (SQP) methods in the numerical optimization literature, which typically exhibit superior convergence properties (Gill et al., 2005; Nocedal & Wright, 2006). Compared to penalty methods, SQP does not regularize the loss in the search direction computation, so it preserves the problem structure (especially important for PDE and control problems) and does not suffer from ill-conditioning issues. Compared to augmented Lagrangian methods, SQP is more

---

[1]Consider $\min x$, s.t. $x = 0$. The soft penalized loss $x + \mu x^2$ leads to the solution $x^\star = -0.5/\mu$, which cannot equal 0 for any finite $\mu > 0$.

[2]Penalty methods differ from vanilla PINN in that they gradually increase the penalty coefficient as needed.

robust to dual initialization, exhibits much faster convergence (quadratic/superlinear for SQP while linear for augmented Lagrangian), and produces solutions with fewer objective function and gradient evaluations (Curtis et al., 2014; Dener et al., 2020; Hong et al., 2023). These promising advantages motivate our study in this paper — design an SQP-based method to leverage recent developments of PINNs for learning PDEs.

In particular, we develop a trust-region SQP method called **trSQP-PINN**. Following the literature Nandwani et al. (2019); Dener et al. (2020); Lu et al. (2021), we first formulate the PDE problem as a hard-constrained problem, where we minimize the empirical loss subject to nonlinear PDE constraints. Then, we apply a trust-region SQP method to impose hard constraints. At each step, we perform a quadratic approximation to the empirical loss and a linear approximation to the constraints. Inspired by the local natural of the approximation, we additionally introduce a trust-region constraint to the parameters. We only trust the approximation and make updates within the trust region. Another critical element of the method is the merit function, which is a scalar-valued function that indicates whether a new iterate is better or worse than the current iterate, in the sense of reducing optimality and feasibility errors. We employ the soft-constrained loss as our merit function. The trust-region radius selection and the step rejection mechanism are both based on the ratio between the actual reduction and the predicted reduction in the soft-constrained loss. If the ratio is large, indicating that our approximation is reliable, we accept the step and enlarge the radius; otherwise, we reject the step and reduce the radius.

A major difference of trSQP-PINN compared to existing penalty and augmented Lagrangian methods is that the trust-region, linear-quadratic subproblem does not rely on any penalty coefficient, effectively avoiding the ill-conditioning issue. The soft-penalized loss is only used to determine whether or not accept the new update. Two *potential* computational bottlenecks when applying our method are (i) solving the trust-region subproblem, and (ii) obtaining the Hessian information of neural network models. For (i), we do not solve the trust-region subproblem exactly (which is still much easier than the ones in penalty and augmented Lagrangian methods), but instead obtain a point that is better than gradient descent (i.e. satisfying the *Cauchy reduction*). For (ii), we employ quasi-Newton updates such as SR1 and (damped) BFGS to approximate the Hessian information.

To further enhance training efficiency and reduce data intensity, we introduce a simple pretraining step focused solely on minimizing the feasibility error. Throughout extensive experiments, we demonstrate that trSQP-PINN significantly outperforms existing hard-constrained methods, improving the accuracy of learned PDE solutions by 1-3 orders of magnitude. Moreover, the improvement of our method is robust against both problem-specific parameters and algorithm tuning parameters, while our pretraining step is broadly effective for other methods as well.

## 2 FROM PINN METHODS TO HARD-CONSTRAINED METHODS

In this section, we first introduce the PDE problem setup and the soft-constrained PINN method. Then, we provide a brief overview of two hard-constrained methods: penalty method and augmented Lagrangian method, and illustrate the motivation of designing a trust-region SQP method.

**Problem setup and PINN method.** We consider an abstraction of the PDE problem with boundary conditions (BCs) and initial conditions (ICs) as follows:

$$\mathcal{F}(u(x,t)) = 0, \quad (x,t) \in \Omega \times \mathcal{T} \subseteq \mathbb{R}^d \times \mathbb{R}^+, \tag{1a}$$

$$\mathcal{B}(u(x,t)) = 0, \quad (x,t) \in \partial\Omega \times \mathcal{T}, \tag{1b}$$

$$\mathcal{I}(u(x,0)) = 0, \quad x \in \Omega. \tag{1c}$$

Here, $(x,t)$ denotes spatial-temporal coordinates with domain $\Omega \times \mathcal{T}$, $\mathcal{F}$ denotes a differential operator that can include multiple PDEs $\{\mathcal{F}_1, \ldots, \mathcal{F}_n\}$, $\mathcal{B}$ is a general form of a boundary-condition operator, and $\mathcal{I}$ is an initial-condition operator. In the context of PDEs, $\mathcal{F}$ may be classified into parabolic, hyperbolic, or elliptic differential operator, and its solution $u(x,t)$ models the change in physical quantities over space and time. For many problems, the analytical solution is not accessible, and solving (1) relies on classical numerical solvers such as finite differences or finite elements.

The recent data-driven approach, PINN, has become popular due to its mesh-free nature and powerful automatic differentiation techniques. In PINN, we apply neural networks to parameterize the solution

$$u_\theta(x,t) = u(x,t),$$

where $(x, t)$ are the NN inputs and $u_\theta(x, t)$ is the output, with $\theta \in \mathbb{R}^p$ representing the NN parameters. Let us denote the labeled observations as $\{(x_i, t_i, u_i)\}_{i=1}^N$, so that we can define the empirical loss:

$$\ell(\theta) = \frac{1}{N} \sum_{i=1}^N (u_i - u_\theta(x_i, t_i))^2 . \tag{2}$$

To enforce constraints (1), we also randomly sample three sets of *unlabeled* points in the spatiotemporal domain $\Omega \times \mathcal{T}$: $\{(x_i^{\mathrm{pde}}, t_i^{\mathrm{pde}})\}_{i=1}^{M_{\mathrm{pde}}}$ for PDE constraints (1a), $\{(x_i^{\mathrm{BC}}, t_i^{\mathrm{BC}})\}_{i=1}^{M_{\mathrm{BC}}}$ for BC constraints (1b), and $\{(x_i^{\mathrm{IC}}, 0)\}_{i=1}^{M_{\mathrm{IC}}}$ for IC constraints (1c). Then, we define the constraint function as

$$c(\theta) = \left( \{\mathcal{F}(u_\theta(x_i^{\mathrm{pde}}, t_i^{\mathrm{pde}}))\}_i; \ \{\mathcal{B}(u_\theta(x_i^{\mathrm{BC}}, t_i^{\mathrm{BC}}))\}_i; \ \{\mathcal{I}(u_\theta(x_i^{\mathrm{IC}}, 0))\}_i \right) \in \mathbb{R}^{M_{\mathrm{pde}} + M_{\mathrm{BC}} + M_{\mathrm{IC}} =: M} . \tag{3}$$

With the above setup, PINN solves Problem (1) by incorporating (3) into (2) as soft constraints:

$$\min_\theta : \ \ell(\theta) + \mu \|c(\theta)\|^2, \tag{4}$$

and applies (unconstrained) gradient-based methods to obtain the solution. Here, $\mu > 0$ is the penalty coefficient that balances between the empirical loss and the constraints. When $\mu$ is large, we penalize the constraint violations severely, thereby ensuring the feasibility of the solution. However, $\mu \to \infty$ also makes the optimization problem (4) ill-conditioned and difficult to converge to a minimum (Krishnapriyan et al., 2021). On the other hand, if $\mu$ is small, then the obtained solution will not satisfy the considered PDEs, making it invalid and less useful in general. Our findings suggest that PINN struggles with PDEs with high coefficients. See Appendix C for experimental results. It is worth mentioning that a more flexible formulation is to use different penalty coefficients for PDEs $\mathcal{F}$, BCs $\mathcal{B}$, and ICs $\mathcal{I}$, while for simplicity, we unify them as $\mu$ in this paper.

**Hard-constrained methods.** As introduced in Section 1, researchers have developed different hard-constrained methods to address the failure modes of PINNs, including penalty methods and augmented Lagrangian methods (Nandwani et al., 2019; Dener et al., 2020; Lu et al., 2021). For this type of methods, we formulate Problem (1) as a constrained nonlinear optimization problem

$$\min_\theta \ \ell(\theta) \quad \text{s.t.} \ c(\theta) = 0, \tag{5}$$

and have the following updating schemes (The pseudocodes are provided in Appendix A).

• **Penalty methods.** Given $(\theta_k, \mu_k)$, we solve $\theta_{k+1} = \arg\min_\theta \ell(\theta) + \mu_k \|c(\theta)\|^2$ with warm initialization at $\theta_k$, and then update $\mu_{k+1} = \rho \mu_k$ with a scalar $\rho > 1$. Unlike the approach of PINN, penalty methods increase the penalty coefficient $\mu$ gradually and solve the subproblem with warm initializations. However, similar to PINNs, a large $\mu$ leads to an ill-conditioned subproblem and slow convergence, and the gradient-based methods may get stuck at poor local minima. This phenomenon is observed in our experiments in Section 4 as well as in Lu et al. (2021).

• **Augmented Lagrangian methods.** Given a triple $(\theta_k, \lambda_k, \mu_k)$, we solve $\theta_{k+1} = \arg\min_\theta \ell(\theta) + \lambda_k^T c(\theta) + \mu_k \|c(\theta)\|^2$ with warm initialization at $\theta_k$, and then update $\lambda_{k+1} = \lambda_k + \mu_k c(\theta_k)$ and $\mu_{k+1} \geq \mu_k$. Compared to penalty methods, $\mu_k$ here is not necessarily increased to infinity to converge to a feasible solution. We can determine whether to increase $\mu_k$ based on the feasibility error $\|c(\theta_k)\|$. For example, we let $\mu_{k+1} = \rho \mu_k$ if the reduction in $\|c(\theta_k)\|$ is insufficient.

• **Motivation of Sequential Quadratic Programming (SQP) methods.** The above hard-constrained methods, though effective for some cases, have a few limitations. First, they have complex nonlinear subproblems involving differential operators, and their quadratic term $\|c(\theta)\|^2$ can destroy the problem structure if any. Second, their convergence is slow. Augmented Lagrangian methods are faster than penalty methods, but only exhibit local linear convergence (Rockafellar, 2022). In addition, it is observed that augmented Lagrangian methods require a significant number of steps to get into the local region and are sensitive to dual initialization (Curtis et al., 2014). In contrast, SQP is one of the most effective methods for both small and large constrained problems. The subproblem of SQP is an inequality-constrained linear-quadratic program, which can be efficiently solved by numerous solvers (notably, we only have to solve it approximately). The convergence rate of SQP matches that of Newton method, and is superlinear if the Hessian is approximated by quasi-Newton update and quadratic if the Hessian is exact (Nocedal & Wright, 2006, Chapters 17 and 18). Most importantly, recent ML techniques, such as randomization, sketching, and subsampling, have been introduced into SQP schemes, enabling further reduction of the computational cost of this method (Hong et al., 2023; Na et al., 2022; Berahas et al., 2021), and making it attractive to investigate in modern SciML problems.

## 3 TRUST-REGION SEQUENTIAL QUADRATIC PROGRAMMING–PINN

In this section, we introduce **trSQP-PINN**, the first SQP-based method designed for solving PDE problems in machine learning. The method has three steps: (i) obtain a linear-quadratic approximation, (ii) incorporate a trust-region constraint, and (iii) update the iterate and trust-region radius. We also introduce a pretraining step to further enhance the training efficiency that is also effective for other methods.

***Step 1: linear-quadratic approximation of Problem*** (5). Let us define the Lagrangian function of (5) as $\mathcal{L}(\theta, \lambda) = \ell(\theta) + \lambda^T c(\theta)$. Given a primal-dual pair $(\theta_k, \lambda_k)$, we denote $\nabla \mathcal{L}_k = \nabla \mathcal{L}(\theta_k, \lambda_k)$ and $c_k = c(\theta_k)$ (similar for $\nabla c_k$, $\nabla \ell_k$, etc.). Then, we construct a linear-quadratic approximation of the nonlinear problem (5) at $(\theta_k, \lambda_k)$ as:

$$\min_{\Delta\theta_k \in \mathbb{R}^p} \quad \nabla \ell_k^T \Delta\theta_k + \frac{1}{2}\Delta\theta_k^T H_k \Delta\theta_k, \tag{6a}$$

$$\text{s.t.} \quad c_k + \nabla c_k \Delta\theta_k = 0, \tag{6b}$$

where (6a) is a quadratic approximation of the empirical loss $\ell(\theta)$ and (6b) is a linear approximation of PDE constraints $c(\theta)$. By using second-order methods, it is not surprising that the Hessian information in $H_k$ enables more significant per-iteration progress compared to first-order methods and better escape from stationary points. Furthermore, the computation (6) does not involve any penalty coefficient, thereby avoiding ill-conditioning issues.

However, when applied to PDE problems, the major concern is obtaining the Hessian matrix $H_k$. In SQP methods, $H_k$ corresponds to the Lagrangian Hessian $\nabla^2_\theta \mathcal{L}_k$, instead of the objective Hessian $\nabla^2 \ell_k$, to leverage curvature information of constraints. This even complicates the computation of $H_k$ due to the differential operators in PDE constraints. To get rid of second-order quantities and boil down to first-order information, we perform quasi-Newton updates for $H_k$. We consider two schemes.

• **Damped BFGS.** Let $s_k := \theta_k - \theta_{k-1}$ and $y_k := \nabla_\theta \mathcal{L}_k - \nabla_\theta \mathcal{L}_{k-1}$. Both vanilla and limited-memory BFGS methods require a critical curvature condition $s_k^T y_k > 0$, which cannot hold in our case due to the saddle structure of the Lagrangian function. Thus, we consider a *damped* version of BFGS. In particular, for a scalar $\delta \in (0, 1)$, we define

$$r_k = \gamma_k y_k + (1 - \gamma_k)H_{k-1}s_k \qquad \text{with} \qquad \gamma_k = \begin{cases} 1 & \text{if } s_k^T y_k \geq \delta s_k^T H_{k-1} s_k, \\ \frac{(1-\delta)s_k^T H_{k-1} s_k}{s_k^T H_{k-1} s_k - s_k^T y_k} & \text{otherwise.} \end{cases}$$

Then, we compute $H_k$ as

$$H_k = H_{k-1} - \frac{H_{k-1}s_k s_k^T H_{k-1}}{s_k^T H_{k-1} s_k} + \frac{r_k r_k^T}{s_k^T r_k}.$$

When $\gamma_k = 1$, the above updating scheme reduces to vanilla BFGS. When $\gamma_k = 0$, $H_k = H_{k-1}$. We note that even if the curvature condition $s_k^T y_k > 0$ may not hold, we ensure $s_k^T r_k \geq \delta s_k^T H_{k-1} s_k > 0$, provided $H_{k-1}$ is positive definite (usually, $H_0 = I$). Common choices of $\delta$ in numerical optimization include $\delta = 0.2$ (Powell, 1978; Goldfarb et al., 2020) and $\delta = 0.25$ (Wang et al., 2017).

• **SR1.** Despite ensuring a modified curvature condition $s_k^T r_k > 0$, the fatal drawback of Damped BFGS is that it essentially employs a positive definite Hessian approximation ($H_{k-1} \succ 0 \Rightarrow H_k \succ 0$). However, the exact Hessian $\nabla^2_\theta \mathcal{L}_k$ cannot be positive definite when training deep neural networks, leading to inaccuracies in approximation. Thus, we also perform the SR1 method, which is preferred when $\nabla^2_\theta \mathcal{L}_k$ lacks positive definiteness, as validated by our experiments in Appendix F. We have

$$H_k = H_{k-1} + \frac{(y_k - H_{k-1}s_k)(y_k - H_{k-1}s_k)^T}{(y_k - H_{k-1}s_k)^T s_k}.$$

The above scheme does not lead to $H_k \succ 0$ when $s_k^T y_k < s_k^T H_{k-1} s_k$.

***Step 2: trust-region constraint.*** The linear-quadratic approximation (6) is intended to be of only local interest. This motivates us to restrict $\theta_k + \Delta\theta_k$ to remain within a trust region centered at $\theta_k$; that is, we incorporate the constraint (6b) along with an additional trust-region constraint ($\Delta_k$ is radius)

$$\|\Delta\theta_k\| \leq \Delta_k. \tag{7}$$

Constraint (7) offers several advantages. First, without (7), the subproblem (6) is well-defined only if $H_k$ is positive definite in the null space $\text{Null}(\nabla c_k)$. This requires further regularizations of the quasi-

Newton updates that cannot preserve the curvature information. Such regularizations are suppressed by (7). Second, (7) serves as a hard regularization on parameters to prevent $\theta_k$ from moving to a worse point. As seen in **Step 3**, we may even skip updating $\theta_k$ if the new point is worse. Additionally, in PDE problems, the Jacobian $\nabla c_k$ tends to be singular since different (especially close) data points carry overlapping local information of PDE solutions. (7) confines the step within a small, reliable area, preventing excessive deviations from local information and mitigating singularity in Jacobians.

However, constraint (7) may conflict with constraint (6b) and lead to

$$\{\Delta\theta_k : c_k + \nabla c_k \Delta\theta_k = 0\} \cap \{\Delta\theta_k : \|\Delta\theta_k\| \leq \Delta_k\} = \emptyset.$$

To resolve this conflict, we employ the relaxation technique originally proposed in Omojokun (1989). In particular, for a scalar $\nu \in (0,1)$, we adjust (6) in two steps:

$$\min_{\widetilde{\Delta\theta}_k \in \mathbb{R}^p} \quad \|c_k + \nabla c_k \widetilde{\Delta\theta}_k\|^2,$$
$$\text{s.t.} \quad \|\widetilde{\Delta\theta}_k\| \leq \nu\Delta_k,$$

then

$$\min_{\Delta\theta_k \in \mathbb{R}^p} \quad \nabla\ell_k^T \Delta\theta_k + \frac{1}{2}\Delta\theta_k^T H_k \Delta\theta_k,$$
$$\text{s.t.} \quad \nabla c_k \Delta\theta_k = \nabla c_k \widetilde{\Delta\theta}_k,$$
$$\|\Delta\theta_k\| \leq \Delta_k.$$

Intuitively, the solution of the left subproblem $\widetilde{\Delta\theta}_k \in \text{Span}(\nabla c_k^T)$ aims to satisfy the linearized constraint (6b) as much as possible within a shrunk radius of $\nu\Delta_k$, while the remaining radius is used to further reduce the quadratic loss (6a) by solving the right subproblem. Although closed-form solutions of these subproblems may be computable via matrix decomposition (cf. Omojokun (1989), (2.1.9)), we only need approximate solutions that are not worse than the Cauchy step, i.e., the steepest descent step satisfying constraint (7) (cf. Nocedal & Wright (2006), Chapter 4). To achieve this, we employ the dogleg method and projected conjugate gradient method to approximately solve the two subproblems.

***Step 3: iterate and radius update.*** With $\Delta\theta_k$, we then decide whether to accept the new iterate $\theta_k + \Delta\theta_k$, depending on how accurate the model problem (6) approximates the original problem (5) and how much progress the new iterate has made towards a (local) solution of (5). To do this, we leverage a non-smooth, soft-constrained loss, called the *merit function*:

$$\phi_\mu(\theta) = \ell(\theta) + \mu\|c(\theta)\|. \tag{8}$$

Note that the empirical loss $\ell(\theta)$ alone is not suitable to justify the approximation quality since a step that decreases $\ell$ may severely violate the constraint $c$. We define the local model of $\phi_\mu(\theta)$ at $\theta_k$ as

$$q_\mu^k(\Delta\theta) = \ell_k + \nabla\ell_k^T \Delta\theta + \frac{1}{2}\Delta\theta^T H_k \Delta\theta + \mu\|c_k + \nabla c_k \Delta\theta\|.$$

Then, we compute the predicted reduction and actual reduction as follows:

$$\text{Pred}_k := q_\mu^k(0) - q_\mu^k(\Delta\theta_k) \qquad \text{and} \qquad \text{Ared}_k := \phi_\mu(\theta_k) - \phi_\mu(\theta_k + \Delta\theta_k).$$

Finally, we calculate the ratio $\eta_k := \frac{\text{Ared}_k}{\text{Pred}_k}$. For two thresholds $0 < \eta_{\text{low}} < \eta_{\text{upp}} < 1$, if $\eta_k \geq \eta_{\text{upp}}$, then it suggests that our approximation is very accurate; we perform $\theta_{k+1} = \theta_k + \Delta\theta_k$, $\Delta_{k+1} = \rho\Delta_k$ for $\rho > 1$, and $\lambda_{k+1} = \arg\min_\lambda \|\nabla\ell_{k+1} + \nabla c_{k+1}^T \lambda\|^2$. If $\eta_{\text{low}} \leq \eta_k < \eta_{\text{upp}}$, then our approximation is moderately accurate; we update $(\theta_k, \lambda_k)$ but preserve the radius $\Delta_{k+1} = \Delta_k$. If $\eta_k < \eta_{\text{low}}$, we reject the update and let $\theta_{k+1} = \theta_k$, $\lambda_{k+1} = \lambda_k$, and $\Delta_{k+1} = \Delta_k/\rho$.

**Remark 1.** We note that the soft-constrained loss (8) is only used in Step 3 to decide whether to accept the update, while it does not affect the step computation in Step 2; thus, effectively overcomes the ill-conditioning issues in penalty and augmented Lagrangian methods. In fact, the coefficient $\mu$ can be selected adaptively, instead of fine tuning it as a parameter. See Conn et al. (2000) for more details.

***Step 0: a pretraining technique.*** Unlike typical ML problems, we restrict our NN parameters to satisfy PDE constraints. Random initializations often fall away from the constraint manifold, complicating training and the search for the optimal solution of the hard-constrained problem. To address this, we develop a simple but effective pretraining step to first train the network using only the physical domain information, allowing it to initialize closer to the feasible region. We apply L-BFGS to solve

$$\theta_{\text{init}} := \text{argmin}_\theta \|c(\theta)\|^2. \tag{9}$$

The pretraining step addresses the scalability concern of SQP. As observed in our experiments, we can use a different set of unlabeled data in the pretraining step (i.e., construct a different constraint function $\tilde{c}$). Then, $\theta_{\text{init}}$ will significantly reduce the number of PDE constraints required in the trSQP-PINN training. Using (9) for initialization is also beneficial for other hard-constrained methods.

We provide the pseudocode for trSQP-PINN in Appendix A.

## 4 EXPERIMENTAL RESULTS

We implement trSQP-PINN on three PDE systems: transport equation, reaction equation, and reaction-diffusion equation. These systems may have simple analytical solutions or be solved by Fast Fourier Transform method. However, when increasing system coefficients, solving these systems becomes numerically challenging (see Krishnapriyan et al. (2021)). We compare our trSQP method with two popular hard-constrained methods: penalty method and augmented Lagrangian method, to demonstrate the superiority of trSQP. We also vary algorithm tuning parameters to illustrate the robustness of the trSQP improvement. Detailed experiment setups are provided in Appendix B. The results of penalty and augmented Lagrangian methods presented here are the ones coupled with the pretraining step, which significantly improve the ones without pretraining as presented in Appendix D.

### 4.1 TRANSPORT EQUATION

We first consider a linear PDE called transport equation. This equation is commonly used to model phenomena in fluid dynamics and wave mechanics, and is applicable to scenarios like pollutant dispersion in rivers or air:

$$\frac{\partial u}{\partial t} + \beta \frac{\partial u}{\partial x} = 0, \quad u(0,t) = u(2\pi,t), \quad u(x,0) = \sin(x), \quad (x,t) \in \Omega \times \mathcal{T}.$$

Here, $\beta \in \mathbb{R}$ is the transport coefficient. The second equation is a periodic boundary condition, and the third equation is an initial condition. The analytical solution is given by $u(x,t) = \sin(x - \beta t)$.

We vary the coefficient $\beta$ across a wide range of values in $\{\pm 1, \pm 10, \pm 20, \pm 30, \pm 40, \pm 50\}$. While previous experiments in Krishnapriyan et al. (2021) have varied small $\beta$ ranging from $10^{-4}$ to $10^{-1}$, we are particularly interested in large $\beta$ where PINN fails. This exploration allows us to distinguish the performance of different hard-constrained methods in learning numerically challenging PDEs and resolving failure modes of PINN. All three methods (as well as PINN) can precisely learn the solution for $|\beta| \leq 0.1$ (cf. Krishnapriyan et al. (2021), Figure 1a).

We plot the absolute errors and relative errors of the three hard-constrained methods in Figure 1a. From the figure, we observe that although penalty and augmented Lagrangian methods achieve low enough errors when $|\beta|$ is small, trSQP-PINN still decreases the errors by 0.5 to one orders of magnitude. Furthermore, trSQP-PINN distinctly outperforms the penalty and augmented Lagrangian methods when $|\beta| \geq 20$. The performance gap becomes even more pronounced with the increasing problem difficulty, where the latter two methods tend to have high errors while trSQP-PINN improves their errors by as much as one to two orders of magnitude for $|\beta| \geq 30$.

Let us examine $\beta = 30$ more closely. We present the solution heatmaps for the three methods in Figure 2a. From the figure, we clearly see that both the penalty and augmented Lagrangian methods fail to capture the complete solution features, while trSQP-PINN successfully learns the periodic dynamics. More precisely, trSQP-PINN achieves an absolute error of 0.72%, which is almost two orders of magnitude lower than the augmented Lagrangian method at 48.21% and penalty method at 51.57%.

The above results of trSQP-PINN highlight its effectiveness in navigating through complex loss landscapes to reach local optima and in mitigating notorious ill-conditioning issues. Its unique feature, the incorporation of trust-region constraints and second-order approximation, significantly enhances the performance compared to other hard-constrained methods.

### 4.2 REACTION EQUATION

We now consider a semi-linear PDE called reaction equation. This equation is used to describe the temporal dynamics of chemical reaction concentrations and is applicable to pharmacokinetics, such as modeling the concentration of drugs in the bloodstream over time:

$$\frac{\partial u}{\partial t} - \alpha u(1 - u) = 0, \quad u(0,t) = u(2\pi,t), \quad u(x,0) = e^{-\zeta(x-\pi)^2}, \quad (x,t) \in \Omega \times \mathcal{T}.$$

Here, $\alpha \in \mathbb{R}$ is the reaction coefficient and $\zeta \in \mathbb{R}$ is the initial condition coefficient. The analytical solution is highly nonlinear and given by $u(x,t) = u(x,0)e^{\alpha t} / \{u(x,0)e^{\alpha t} + 1 - u(x,0)\}$.

Following the transport equation setup, we fix $\zeta = 2$ and vary the reaction coefficient $\alpha$ in the set $\{\pm 1, \pm 10, \pm 20, \pm 30, \pm 40, \pm 50\}$. We plot the absolute and relative errors of the three hard-constrained

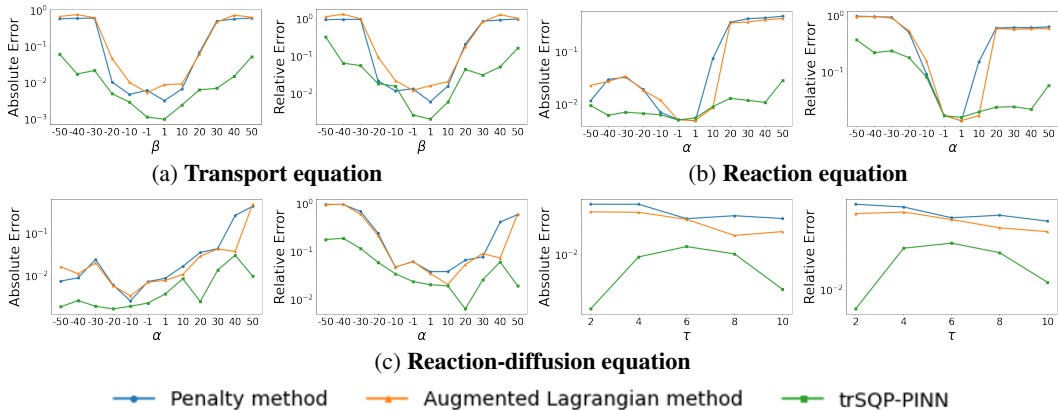

Figure 1: **Absolute and relative errors of three hard-constrained methods for learning PDEs.** *All three methods can learn PDE solutions well when the PDE coefficients $|\beta|$ and $|\alpha|$ are small, with trSQP-PINN yielding the lowest errors in solving transport and reaction-diffusion equations. However, as the coefficients increase and the problems become more challenging, trSQP-PINN significantly outperforms the other methods.*

methods in Figure 1b. While all three methods maintain low errors at small values of $|\alpha|$, as the problem difficulty increases with $|\alpha| \geq 20$, the advantage of trSQP-PINN grows significantly, exhibiting errors that are one to two orders of magnitude lower compared to the other methods. We should mention that the error gaps between trSQP-PINN and the other methods at $\alpha = -50$ to $-20$ are narrower than those at $\alpha = 20$ to $50$. Nevertheless, as displayed in Appendix E, the learned trSQP-PINN solutions are still remarkably superior to the others, and the small error differences are largely attributed to the solution's unique sharp corner point pattern — there is an extreme value *at and only at* $x \approx \pi$ and $t \approx 0$ (meaning most areas in the domain region cannot distinguish between different methods, and only near the point $(x \approx \pi, t \approx 0)$ can the methods be truly tested for their effectiveness).

We take $\alpha = 30$ as an illustrative example and draw the solution heatmaps for the three methods in Figure 2b. From the figure, we see that trSQP-PINN excels in solving the reaction equation with high reaction coefficients. The method precisely captures the initial condition features, periodic boundary features, as well as the sharp transition patterns. In terms of errors, trSQP-PINN achieves a relative error of only 2.52%, which is nearly two orders of magnitude lower than that of the penalty method and augmented Lagrangian method, which have relative errors of 62.62% and 58.32%, respectively.

### 4.3 REACTION-DIFFUSION EQUATION

Finally, we consider a parabolic semi-linear PDE called reaction-diffusion equation, which adds a second-order diffusion term to the reaction equation:

$$\frac{\partial u}{\partial t} - \tau \frac{\partial^2 u}{\partial x^2} - \alpha u(1-u) = 0, \quad u(0,t) = u(2\pi,t), \quad u(x,0) = e^{-\zeta(x-\pi)^2}, \quad (x,t) \in \Omega \times \mathcal{T}.$$

Here, $\alpha \in \mathbb{R}$ and $\tau > 0$ are the reaction and diffusion coefficients, and $\zeta \in \mathbb{R}$ is the initial condition coefficient. The exact solution is solved by Fast Fourier Transform method (Duhamel & Vetterli, 1990).

We vary the reaction coefficient $\alpha$ in $\{\pm 1, \pm 10, \pm 20, \pm 30, \pm 40, \pm 50\}$ while fixing $\tau = 2$ and $\zeta = 2$, and vary the diffusion coefficient $\tau$ in $\{2, 4, 6, 8, 10\}$ while fixing $\alpha = 20$ and $\zeta = 2$. The absolute and relative errors of the three methods are plotted in Figure 1c. While all three methods maintain low errors at small values of $|\alpha|$, trSQP-PINN still achieves lower errors than the other two methods due to its ability to handle the high nonlinearity of the reaction-diffusion equation. Furthermore, as $|\alpha|$ increases, trSQP-PINN significantly outperforms the other two methods. Specifically, despite the increased problem complexity and ill-conditioning issues introduced by the second-order diffusion term, trSQP-PINN achieves an improvement of one to two orders of magnitude in both absolute and relative errors for $\alpha \geq 20$. For $\alpha \leq -20$, trSQP-PINN even reduces the relative error by two to three orders of magnitude. We also observe from Figure 1c that the improvement of trSQP-PINN is robust across different diffusion coefficients $\tau$. For example, even when $\tau$ is as large as 10, only trSQP-PINN achieves both absolute and relative errors below $10^{-2}$.

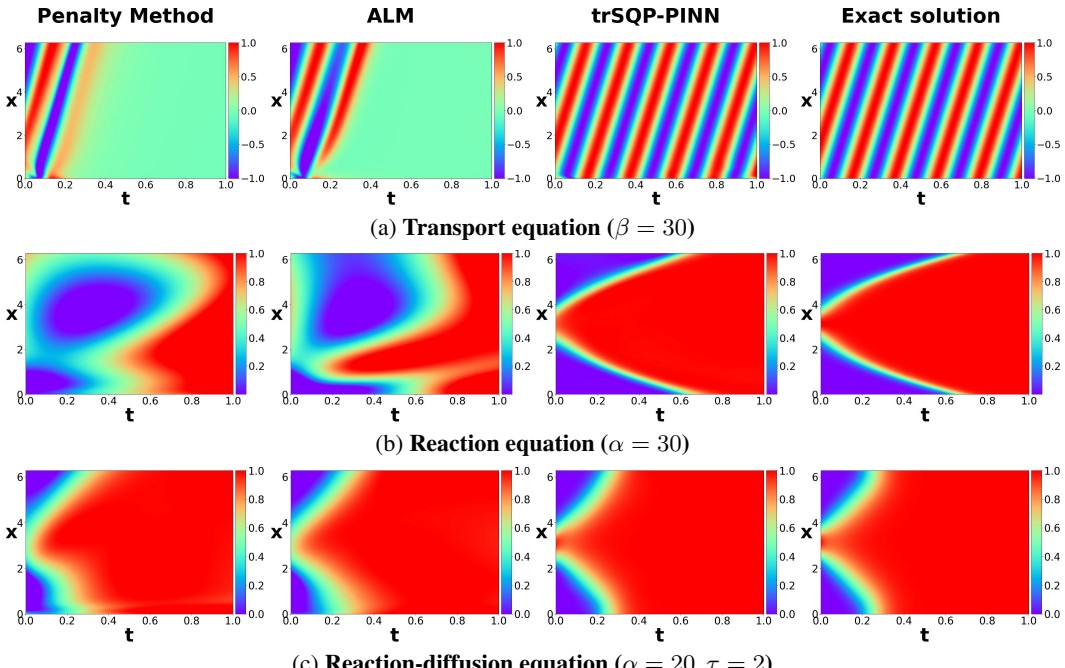

(a) **Transport equation** ($\beta = 30$)

(b) **Reaction equation** ($\alpha = 30$)

(c) **Reaction-diffusion equation** ($\alpha = 20, \tau = 2$)

Figure 2: **Solutions of three hard-constrained methods for learning PDEs.** *TrSQP-PINN can fully recover PDE solutions at high coefficients, including initial conditions, periodic boundary conditions, and sharp transition patterns. The other methods fail to capture the complete solution features.*

We plot the solution heatmaps for $\alpha = 20$ and $\tau = 2$ in Figure 2c. We see that only trSQP-PINN captures the initial condition and periodic boundary features, as well as sharp transition areas. In particular, trSQP-PINN achieves an absolute error of 0.66%, which is one order of magnitude lower than the penalty method at 6.84% and the augmented Lagrangian method at 5.54%.

### 4.4 SENSITIVITY TO TUNING PARAMETERS

We test the sensitivity of trSQP-PINN's performance to various tuning parameters, including the depth and width of the neural networks, the number of pretraining and training data points, and the choice of quasi-Newton updating schemes (damped BFGS v.s. SR1 v.s. identity Hessians). We also implement penalty and augmented Lagrangian methods for comparison. The results of testing the depth and width of the neural networks and the number of pretraining data points are deferred to Appendix F, while the results of testing the number of training data points and the choice of quasi-Newton updates are summarized in Figure 3 and Table 1.

In particular, we vary the number of training data points $N$ in the set $\{100, 200, 500, 1000\}$. From Figure 3, we observe that trSQP-PINN consistently outperforms the penalty and augmented Lagrangian methods by achieving one to two orders of magnitude lower errors. This suggests the robustness of trSQP-PINN's superiority across different sizes of training datasets and highlights its ability to extract more PDE system information from limited data.

We employ either damped BFGS, SR1, or simply the identity matrix to approximate the Lagrangian Hessian in trSQP-PINN, with results presented in Table 1. This experiment focuses exclusively on trSQP-PINN as other methods do not involve estimating the Hessian of the Lagrangian function. As expected in Section 3, SR1 achieves lower errors when learning more complex systems, such as reaction and reaction-diffusion equations, where the exact Hessian lacks positive definiteness. Moreover, when using identity Hessians, trSQP-PINN performs worse but still better than the penalty and augmented Lagrangian methods. This indicates that (i) the use of the trust-region technique can overcome the ill-conditioning issue and better learn PDE solutions, and (ii) quasi-Newton updates to approximate second-order information can significantly boost the performance of trSQP-PINN.

Overall, we observe that the superior performance of trSQP-PINN over other methods remains robust across all tuning parameters. As seen in Appendix F, trSQP-PINN performs reasonably well even with

Table 1: **Different Lagrangian Hessian approximation methods for trSQP-PINN.** *For each problem, the smaller error between different updating schemes is bold. TrSQP-PINN with SR1 generally achieves lower absolute and relative errors than using damped BFGS. Using identity Hessians would restricts the performance of trSQP-PINN.*

| Hessian Estimation Method | Error $(10^{-1})$ | Transport | Reaction | Reaction-diffusion |
|---|---|---|---|---|
| Damped BFGS | Abs_err | **0.043** | 0.132 | 0.056 |
|  | Rel_err | **0.148** | 0.257 | 0.138 |
| **SR1** | Abs_err | 0.072 | **0.121** | **0.026** |
|  | Rel_err | 0.321 | **0.252** | **0.066** |
| Identity | Abs_err | 2.312 | 0.172 | 0.086 |
|  | Rel_err | 5.663 | 0.323 | 0.206 |

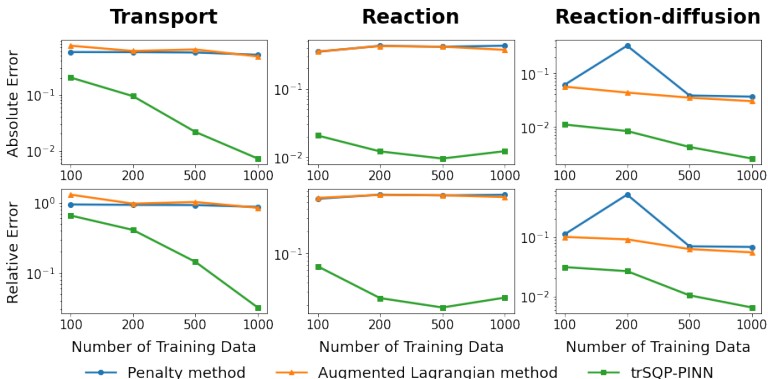

Figure 3: **Absolute and relative errors of three hard-constrained methods with varying number of training data points.** *TrSQP-PINN consistently outperforms the other methods and maintains low errors even if it is trained on a small dataset.*

limited pretraining data (e.g., $M = 30$), indicating its ability to avoid suboptimal solutions resulting from poor initialization. Furthermore, even when reducing the depth and/or width of the neural network, thereby restricting the expressive power of the network, trSQP-PINN still performs robustly. For example, using a single-layer network with 50 neurons or a 4-layer network with 10 neurons per layer, only trSQP-PINN successfully learns the considered PDE systems.

## 5  CONCLUSION AND FUTURE WORK

We designed a direct, hard-constrained deep learning method — trust-region Sequential Quadratic Programming (trSQP-PINN) method — to address the failure modes of PINNs for solving PDE problems. At each step, the method performs a linear-quadratic approximation to the original nonlinear PDE problem, and incorporates an additional trust-region constraint to respect the local nature of the approximation. The method employs quasi-Newton updates to approximate second-order information and leverages a soft-constrained loss to adaptively update the trust-region radius. Compared to popular hard-constrained methods such as penalty methods and augmented Lagrangian methods, our extensive experiments have shown that trSQP-PINN robustly exhibits superior performance over a wide range of PDE coefficients, achieving 1-2 orders of magnitude lower errors. The enhanced performance stems from leveraging trust-region technique and second-order information, helping trSQP-PINN to mitigate ill-conditioning issues and navigate through complex loss landscapes efficiently.

Future work includes exploring more numerical techniques for solving SciML problems and applying SQP-type methods to inverse problems and operators learning problems. For example, we can utilize Randomized Numerical Linear Algebra techniques (e.g., sketching or subsampling) to approximately solve trust-region subproblems to further reduce the computational cost of trSQP. Lu et al. (2021) has demonstrated the effectiveness of penalty and augmented Lagrangian methods for solving inverse problems. Designing a suitable SQP-type method for such cases is also an important open direction.

**Ethics Statement.** This paper does not involve human subjects, nor does it present potentially harmful methodologies or applications that could result in discrimination against different communities.

**Reproducibility.** The detailed experimental setup is presented in Appendix B, and the source code is provided in the supplementary material.

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

# Supplementary material: Physics-Informed Neural Networks with Trust-Region Sequential Quadratic Programming

## A  PSEUDOCODES FOR PENALTY, AUGMENTED LAGRANGIAN, AND TRSQP METHODS

We define $\phi_\mu^{\mathrm{P}}(\theta) = \ell(\theta) + \mu\|c(\theta)\|^2$ to be the soft-constrained loss for penalty method and $\phi_\mu^{\mathrm{AL}}(\theta, \lambda) = \ell(\theta) + \lambda^T c(\theta) + \mu\|c(\theta)\|^2$ to be the augmented Lagrangian function. Our stopping criterion for solving unconstrained subproblems in penalty method, augmented Lagrangian method, and the pretraining step is set as follows (we replace $\phi_\mu^{\mathrm{P}}$ by $\phi_\mu^{\mathrm{AL}}$ and $c$ for the latter two subproblems):

$$\|\nabla\phi_\mu^{\mathrm{P}}(\theta^l)\|_\infty \leq \text{g\_tol} \qquad \textbf{OR} \qquad \|\theta^{l+1} - \theta^l\| \leq \text{f\_tol} \qquad \textbf{OR} \qquad l \geq l_{\max}, \qquad (A.1)$$

where g_tol and f_tol are the convergence tolerances, $l_{\max}$ is the maximum number of iterations, and $l$ is the inner-loop iteration index for solving the subproblem.

Algorithms 1, 2, and 3 present the pseudocodes for penalty method, augmented Lagrangian method, and trSQP method. In the case of penalty method and augmented Lagrangian method, we adhere to the approach outlined in the literature Lu et al. (2021), where we terminate the algorithm when the penalty coefficient $\mu$ exceeds the threshold $\mu_{\max}$ to alleviate their ill-conditioning issues during training.

---

**Algorithm 1** Penalty Method

---

**Inputs:** $\mu_0 = 1$, $\rho = 1.1$, $\mu_{\max} = \rho^{100}$, g_tol = f_tol = $10^{-9}$, $l_{\max} = 2 \times 10^4$;
1: **(Pretrain)** $\theta_{\text{init}} \leftarrow \arg\min_\theta \|c(\theta)\|^2$: train the model until (A.1) is satisfied;
2: $k \leftarrow 0$ and $\theta_0 \leftarrow \theta_{\text{init}}$;
3: **repeat**
4:     $\theta_{k+1} \leftarrow \arg\min_\theta \phi_{\mu_k}^{\mathrm{P}}(\theta)$ with initialization at $\theta_k$: train the model until (A.1) is satisfied;
5:     $\mu_{k+1} \leftarrow \rho\mu_k$;
6:     $k \leftarrow k + 1$;
7: **until** $\mu_k \geq \mu_{\max}$

---

**Algorithm 2** Augmented Lagrangian Method

---

**Inputs:** $\mu_0 = 1$, $\lambda_0 = 0$, $\rho = 1.1$, $\mu_{\max} = \rho^{100}$, g_tol = f_tol = $10^{-9}$, $l_{\max} = 2 \times 10^4$;
1: **(Pretrain)** $\theta_{\text{init}} \leftarrow \arg\min_\theta \|c(\theta)\|^2$: train the model until (A.1) is satisfied;
2: $k \leftarrow 0$ and $\theta_0 \leftarrow \theta_{\text{init}}$;
3: **repeat**
4:     $\theta_{k+1} \leftarrow \arg\min_\theta \phi_{\mu_k}^{\mathrm{AL}}(\theta, \lambda_k)$ with initialization at $\theta_k$: train the model until (A.1) is satisfied;
5:     $\lambda_{k+1} \leftarrow \lambda_k + \mu_k c_k$;
6:     $\mu_{k+1} \leftarrow \rho\mu_k$;
7:     $k \leftarrow k + 1$;
8: **until** $\mu_k \geq \mu_{\max}$

---

## B  DETAILED EXPERIMENT SETUP

For all three PDE systems (transport, reaction, reaction-diffusion), we define the spatial domain $\Omega = [0, 2\pi]$ and the temporal domain $\mathcal{T} = [0, 1]$. For learning transport and reaction equations, our training dataset consists of $N$ labeled points and $M = M_{\text{pde}} + M_{\text{BC}} + M_{\text{IC}}$ unlabeled points uniformly sampled from $\Omega \times \mathcal{T}$. However, due to the use of the Fast Fourier Transform in obtaining exact solutions of reaction-diffusion equation, it is time-consuming to generate solutions for each pair $(x_i, t_i) \in \Omega \times \mathcal{T}$. Thus, we let $N_{\text{xgrid}}$ and $N_{\text{tgrid}}$ denote the number of evenly distributed points in $\Omega$ and $\mathcal{T}$, respectively, forming an $N_{\text{xgrid}} \times N_{\text{tgrid}}$ grid of points. We then obtain the training dataset for learning reaction-diffusion equation by uniformly sampling from the grid points while still sampling unlabeled points from $\Omega \times \mathcal{T}$. The $N_{\text{xgrid}} \times N_{\text{tgrid}}$ grid is also used for evaluating all the methods. We use $M^{\text{pretrain}}$ to denote the number of unlabeled points for the pretraining phase, which is evenly distributed among $M_{\text{pde}}$, $M_{\text{BC}}$, and $M_{\text{IC}}$. Analogously, we use $M^{\text{train}}$ points for the training phase.

---

**Algorithm 3** Trust-Region Sequential Quadratic Programming Method

---

**Inputs:** $\mu_{-1} = 1$, $H_0 = I$, $\Delta_0 = 1$, $\delta = 0.2$, $\nu = 0.8$, $\eta_{\text{low}} = 10^{-8}$, $\eta_{\text{upp}} = 0.3$, $\rho = 2$, g_tol = f_tol $= 10^{-9}$, $K_{\max} = l_{\max} = 2 \times 10^4$;

1: **(Pretrain)** $\theta_{\text{init}} \leftarrow \arg\min_\theta \|c(\theta)\|^2$: train the model until (A.1) is satisfied;     ▷ **Step 0**
2: $k \leftarrow 0$, $\theta_0 \leftarrow \theta_{\text{init}}$, $\lambda_0 = \arg\min_\lambda \|\nabla\ell_0 + \nabla c_0^T \lambda\|^2$;
3: **while** $k \leq K_{\max}$ **do**
4:     Compute quasi-Newton update for $H_k$ via Damped BFGS (with $\delta$) or SR1;     ▷ **Step 1**
5:     Solve trust-region subproblem for $\widetilde{\Delta\theta}_k$ and then $\Delta\theta_k$ (with $\nu$);     ▷ **Step 2**
6:     Compute

$$\mu_k := \max\left\{\mu_{k-1}, \frac{\nabla\ell_k^T \Delta\theta_k + \frac{1}{2}\Delta\theta_k^T H_k \Delta\theta_k}{0.7(\|c_k\| - \|c_k + \nabla c_k \Delta\theta_k\|)}\right\} \quad \text{and then} \quad \eta_k := \frac{\text{Ared}_k}{\text{Pred}_k};$$

7:     **if** $\eta_k \geq \eta_{\text{upp}}$ **then**     ▷ **Step 3**
8:         $\theta_{k+1} \leftarrow \theta_k + \Delta\theta_k$, $\Delta_{k+1} \leftarrow \rho\Delta_k$, $\lambda_{k+1} = \arg\min_\lambda \|\nabla\ell_{k+1} + \nabla c_{k+1}^T \lambda\|^2$;
9:     **else if** $\eta_{\text{low}} \leq \eta_k < \eta_{\text{upp}}$ **then**
10:         $\theta_{k+1} \leftarrow \theta_k + \Delta\theta_k$, $\Delta_{k+1} \leftarrow \Delta_k$, $\lambda_{k+1} = \arg\min_\lambda \|\nabla\ell_{k+1} + \nabla c_{k+1}^T \lambda\|^2$;
11:     **else**
12:         $\theta_{k+1} \leftarrow \theta_k$, $\Delta_{k+1} \leftarrow \Delta_k/\rho$, $\lambda_{k+1} \leftarrow \lambda_k$;
13:     **end if**
14:     **if** $\eta_k < \eta_{\text{low}}$ **AND** ($\|\theta_{k+1} - \theta_k\| \leq$ f_tol **OR** $\|\nabla_\theta \mathcal{L}_{k+1}\|_\infty \leq$ g_tol) **then**
15:         **STOP;**
16:     **end if**
17:     $k \leftarrow k + 1$;
18: **end while**

---

Given a spatial-temporal pair $(x_i, t_i)$ on the grid, the observation (i.e., label) is constructed as $u_i = u(x_i, t_i) + \epsilon_i$, where $u(\cdot, \cdot)$ is either an analytical PDE solution (transport, reaction) or a solution approximation derived by Fast Fourier Transform (reaction-diffusion), and $\epsilon_i$ is a random noise term added to enhance generalization. The prediction accuracy is assessed using absolute errors and relative errors, defined as

$$\text{Abs\_err} := \frac{1}{N_{\text{xgrid}} \cdot N_{\text{tgrid}}} \sum_{i=1}^{N_{\text{xgrid}}} \sum_{j=1}^{N_{\text{tgrid}}} \|u_\theta(x_i, t_j) - u(x_i, t_j)\|_2,$$

$$\text{Rel\_err} := \frac{1}{N_{\text{xgrid}} \cdot N_{\text{tgrid}}} \sum_{i=1}^{N_{\text{xgrid}}} \sum_{j=1}^{N_{\text{tgrid}}} \frac{\|u_\theta(x_i, t_j) - u(x_i, t_j)\|_2}{\|u(x_i, t_j)\|_2}.$$

We employ a 4-layer, fully-connected neural network with $50$ neurons in each layer. The activation function is tanh. The subproblems in penalty and augmented Lagrangian methods are solved by L-BFGS coupled with backtracking line search. We utilize Flax to build the neural network and Jax to perform automatic differentiation.

For both penalty method and augmented Lagrangian method, the initial penalty coefficient $\mu_0$ is set to 1 with an increasing factor $\rho = 1.1$. The initial dual vector is set to $\lambda_0 = 0$. For trSQP-PINN, the initial penalty coefficient $\mu_{-1}$ is set to 1 and adaptively selected at each step. The initial trust-region radius $\Delta_0$ is set to 1. We apply SR1 for quasi-Newton update in the method.

We employ a fine grid for evaluation, with $N_{\text{xgrid}} = 2560$, $N_{\text{tgrid}} = 1000$, and $M^{\text{pretrain}} = 150$. We set $M^{\text{train}} = 12$ for transport equation and $M^{\text{train}} = 7$ for both reaction and reaction-diffusion equations. The use of different numbers of unlabeled points for learning different PDEs is largely due to varying properties of the PDEs; we prefer fewer unlabeled data points when the PDEs become more complex and make the loss landscape more tortuous. For all three PDE systems, we set $N = 1000$. We have tested the robustness of our method against $N$ and $M^{\text{pretrain}}$ in Appendix F.

Table 2: **Absolute and relative errors of PINN method with varying penalty coefficients.** *PINN exhibits high prediction errors, which increase as the penalty coefficient $\mu$ increases, indicating worsening ill-conditioning issues. The errors in predicting the solutions of the transport equation peak at $\mu = 1000$, while PINN consistently performs poorly in predicting the solutions of the reaction and reaction-diffusion equations due to severe ill-conditioning already occurring at $\mu = 1$.*

| Coefficient | Error $(10^{-1})$ | Transport | Reaction | Reaction-diffusion |
|:---:|:---:|:---:|:---:|:---:|
| $\mu = 1$ | Abs_err | 5.746 | 7.806 | 8.954 |
|  | Rel_err | 9.403 | 9.999 | 9.999 |
| $\mu = 10$ | Abs_err | 5.912 | 7.806 | 8.954 |
|  | Rel_err | 9.675 | 10.000 | 10.000 |
| $\mu = 100$ | Abs_err | 7.655 | 7.806 | 8.954 |
|  | Rel_err | 13.808 | 10.000 | 10.000 |
| $\mu = 1000$ | Abs_err | 11.329 | 7.806 | 8.953 |
|  | Rel_err | 19.176 | 9.999 | 9.999 |

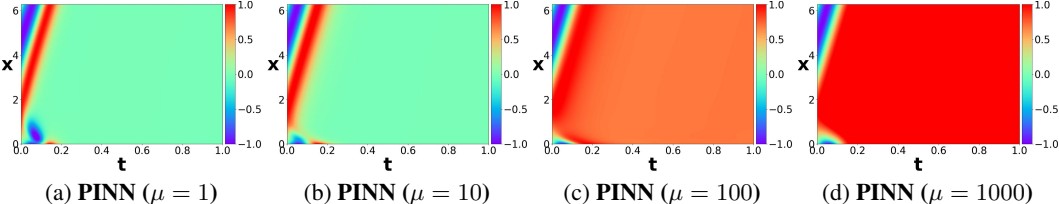

(a) **PINN** ($\mu = 1$)     (b) **PINN** ($\mu = 10$)     (c) **PINN** ($\mu = 100$)     (d) **PINN** ($\mu = 1000$)

Figure 4: **PINN solutions for learning transport equation.** *PINN cannot recover the solution of the transport equation regardless of the penalty coefficient $\mu$ (see Figure 2a for exact solution) and the prediction gets worse as $\mu$ increases.*

## C  FAILURE MODES OF PINN METHOD

In this section, we implement vanilla PINN method outlined in Section 2 (see (4)) with varying penalty coefficient $\mu$. We illustrate that PINN fails to recover the solutions of all three systems with large system coefficients.

We set $\beta = 30$ for transport equation, $\alpha = 30$ and $\zeta = 2$ for reaction equation, and $\alpha = 20$, $\tau = 2$, and $\zeta = 2$ for reaction-diffusion equation. Our results are summarized in Table 2. Comparing this table with Figure 1, we observe that PINN performs much worse than the three hard-constrained methods. Moreover, as the coefficient $\mu$ increases, the error explodes due to the worsening ill-conditioning issue. To visualize the error table, we take the transport equation as an example and draw the heatmap of the PINN solution in Figure 4. The heatmap of the exact solution is shown in Figure 2a. From these two figures, we clearly see that PINN with varying coefficient $\mu$ fails to learn the solution of the transport equation. We have similar observations for the reaction and reaction-diffusion equations. This observation is consistent with the recent study (Krishnapriyan et al., 2021).

## D  PRETRAINING V.S. NO PRETRAINING

In this section, we demonstrate that our pretraining step is generally beneficial for both penalty method and augmented Lagrangian method. To this end, we compare the results of these two methods with and without pretraining. For the methods without pretraining, we employ random initializations. The coefficients of the three PDE problems are set as described in Appendix C.

Our results are summarized in Table 3. From the table, it is evident that both penalty and augmented Lagrangian methods without pretraining perform significantly worse than their pretrained counterparts when solving reaction and reaction-diffusion equations. While the unpretrained penalty method may exhibit similar performance to its pretrained counterpart for solving transport equation, we find that the

Table 3: **Absolute and relative errors of penalty and augmented Lagrangian methods with/without pretraining.** *For each method, the smaller error between pretraining and unpretraining is bold. The unpretrained versions of penalty and augmented Lagrangian methods generally perform much worse than their pretrained counterparts in learning three PDEs. One exception is about the penalty method for solving transport equation. The pretrained and unpretrained methods show similar results.*

| PDE | Error $(10^{-1})$ | Penalty | | Augmented Lagrangian | |
|---|---|---|---|---|---|
| | | Unpretrained | Pretrained | Unpretrained | Pretrained |
| Transport | Abs_err | 5.264 | **5.157** | 9.860 | **4.821** |
| | Rel_err | 8.986 | **8.746** | 17.313 | **8.479** |
| Reaction | Abs_err | 8.953 | **4.339** | 7.808 | **3.766** |
| | Rel_err | 9.999 | **6.262** | 10.003 | **5.832** |
| Reaction-diffusion | Abs_err | 8.953 | **0.364** | 9.754 | **0.300** |
| | Rel_err | 9.999 | **0.684** | 12.495 | **0.554** |

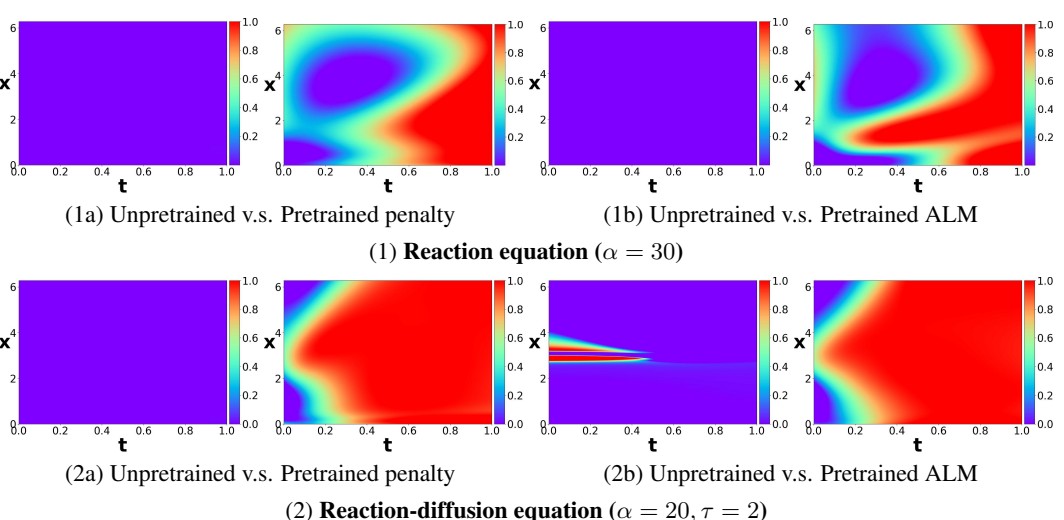

(1a) Unpretrained v.s. Pretrained penalty (1b) Unpretrained v.s. Pretrained ALM

(1) **Reaction equation ($\alpha = 30$)**

(2a) Unpretrained v.s. Pretrained penalty (2b) Unpretrained v.s. Pretrained ALM

(2) **Reaction-diffusion equation ($\alpha = 20, \tau = 2$)**

Figure 5: **Solutions of penalty and augmented Lagrangian methods with/without pretraining for learning reaction and reaction-diffusion equations.** *For both methods, the predictions without pretraining (the left panels) are significantly worse than the predictions with pretraining (the right panels).*

unpretrained augmented Lagrangian method is markedly less effective than the pretrained one. Overall, our pretraining step proves to be effective for hard-constrained methods, improving the prediction error by as much as an order of magnitude.

In Figure 5, we visualize the error table by drawing the solution heatmaps of penalty and augmented Lagrangian methods for learning reaction and reaction-diffusion equations. From the figure, we observe that unpretrained methods fail to capture any solution features, while pretrained methods perform reasonably better (especially for augmented Lagrangian). Thus, we conclude that our pretraining phase sets a robust baseline for further optimization and enhances the efficacy of hard-constrained methods in finding better solutions.

# E  ADDITIONAL RESULTS FOR REACTION EQUATION

In this section, we present additional solution heatmaps for learning the reaction equation with $\alpha \in \{-20, -30, -40, -50\}$ in Figure 6. As explained in Section 4.2, although the errors among the three methods are close, Figure 6 reveals significant differences in their learned solutions. Learning the reaction equation with a negative coefficient is extremely difficult near the point ($x \approx \pi, t \approx 0$), and only

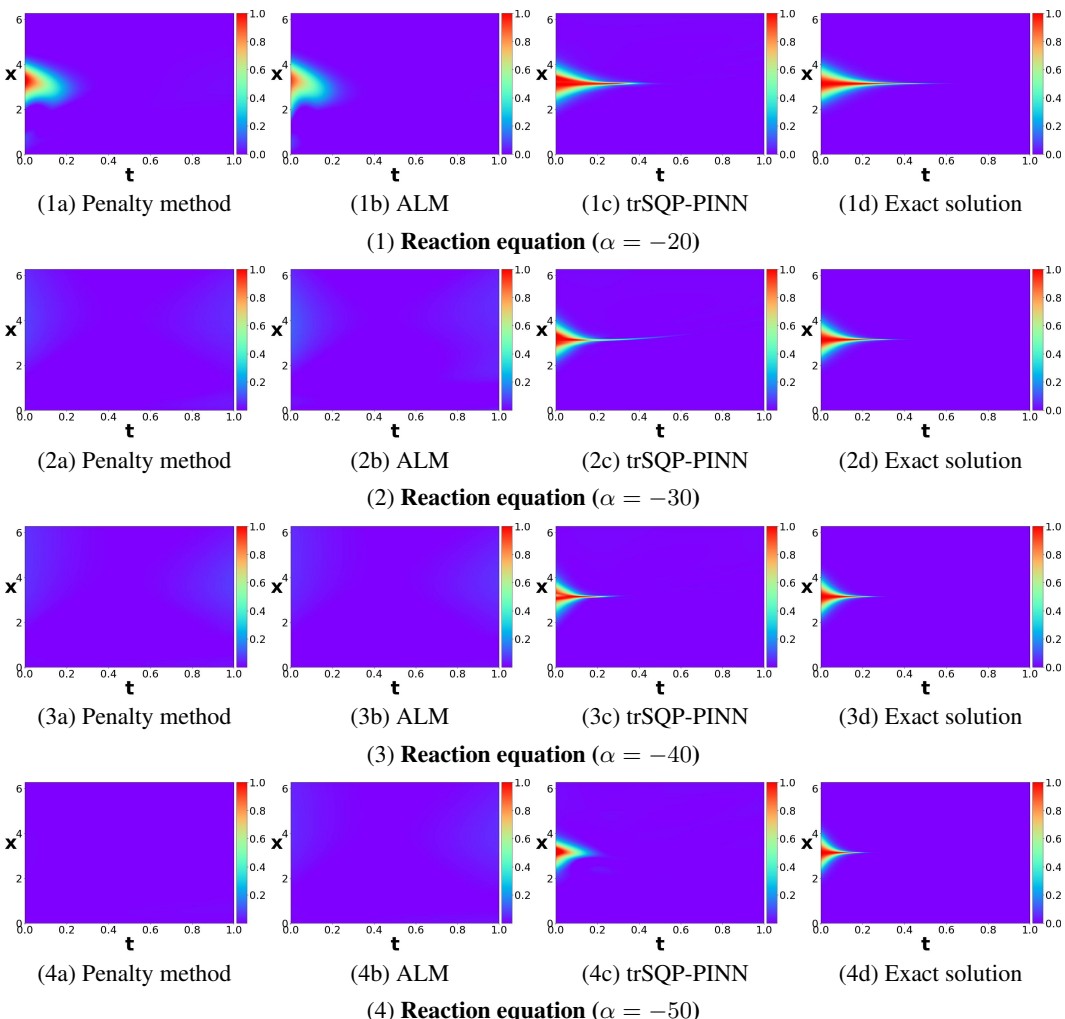

Figure 6: **Solutions of three hard-constrained methods for learning reaction equation.** *Only trSQP-PINN can reasonably recover the solution of the reaction equation and performs significantly better than the other methods at $\alpha \leq -20$.*

trSQP-PINN can reasonably recover the unique sharp corner point pattern for $\alpha \leq -20$. The small differences in errors are also attributed to such an unique pattern in the solutions; most areas in the domain region cannot distinguish between different methods, and only near the point $(x \approx \pi, t \approx 0)$ can the methods be truly tested for their effectiveness

## F    ADDITIONAL RESULTS FOR SENSITIVITY EXPERIMENT

In conjunction with the sensitivity experiments described in Section 4.4, we present in this section additional sensitivity experiments to test the robustness of trSQP-PINN's performance against some other tuning parameters, including the depth and width of the neural networks and the number of pretraining data points. We also include penalty and augmented Lagrangian methods for comparison. For each experiment, only one parameter is altered from the default settings detailed in Appendix B. The problem coefficients are provided in Appendix C; that is, $\beta = 30$ for transport equation, $\alpha = 30$ and $\zeta = 2$ for reaction equation, and $\alpha = 20, \tau = 2$, and $\zeta = 2$ for reaction-diffusion equation.

Figure 7 shows the sensitivity results for varying the number of pretraining data points. We vary $M^{\text{pretrain}}$ in the sets $\{30, 45, 150, 300\}$. As discussed in Section 3, having fewer pretraining data results

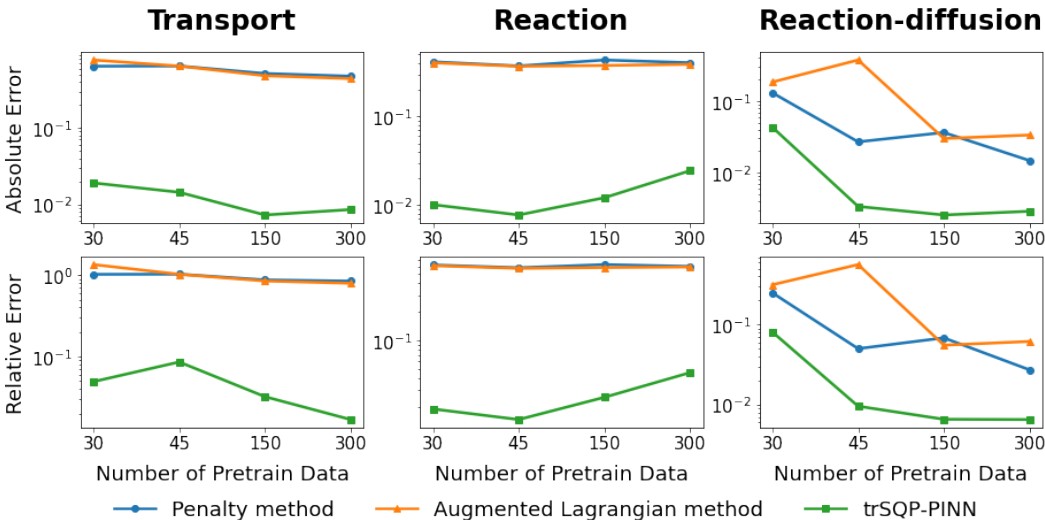

Figure 7: **Absolute and relative errors of three hard-constrained methods with varying number of pretraining data points.** *TrSQP-PINN consistently outperforms the other methods and maintains low errors even if it is initialized far from the feasible region.*

in initializations far from the feasible region. The figure demonstrates that trSQP-PINN is significantly more robust to such initializations compared to the other hard-constrained methods.

Figures 8 and 9 show the sensitivity results for varying the depth and width of the neural networks. We vary the depth in the set $\{1, 2, 3, 4\}$ (while fixing the width 50) and vary the width in the set $\{10, 20, 30, 40, 50\}$ (while fixing the depth 4). Our results suggest that only trSQP-PINN can consistently learn PDE solutions with relatively small networks. From Figure 8, we see that trSQP-PINN achieves lowest errors when the depth decreases to one layer across all three problems; however, we also see that the errors double compared to using 4-layer networks. This increase in errors can be attributed to the limited expressive power of these shallow networks. From Figure 9, trSQP-PINN also consistently outperforms the other methods across all neural network widths. However, when solving the transport and reaction equations, all methods exhibit high errors with only 10 neurons per layer, suggesting low expressivity of the neural networks. When solving the reaction-diffusion equation, trSQP-PINN is able to maintain low errors across all neural network widths, while the penalty and augmented Lagrangian methods are highly unstable, showing very high errors when the neural network width is 20 or 30.

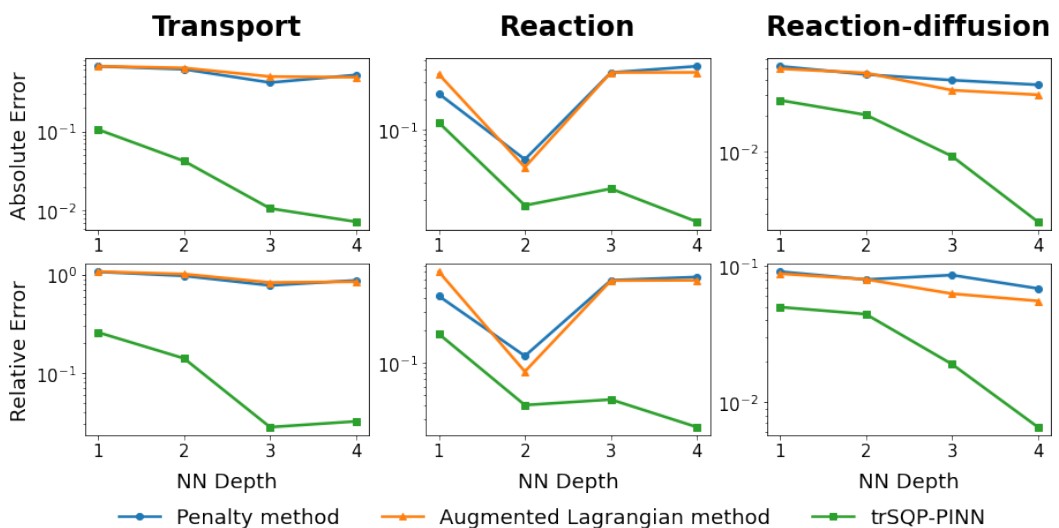

Figure 8: **Absolute and relative errors of three hard-constrained methods with varying neural networks depth.** *TrSQP-PINN consistently outperforms the other methods and maintains low errors even when trained on shallower neural networks. However, while trSQP-PINN achieves the lowest errors on shallow networks, the errors double compared to those on 4-layer networks. This suggests that sufficient network expressivity is necessary for better learning of PDEs.*

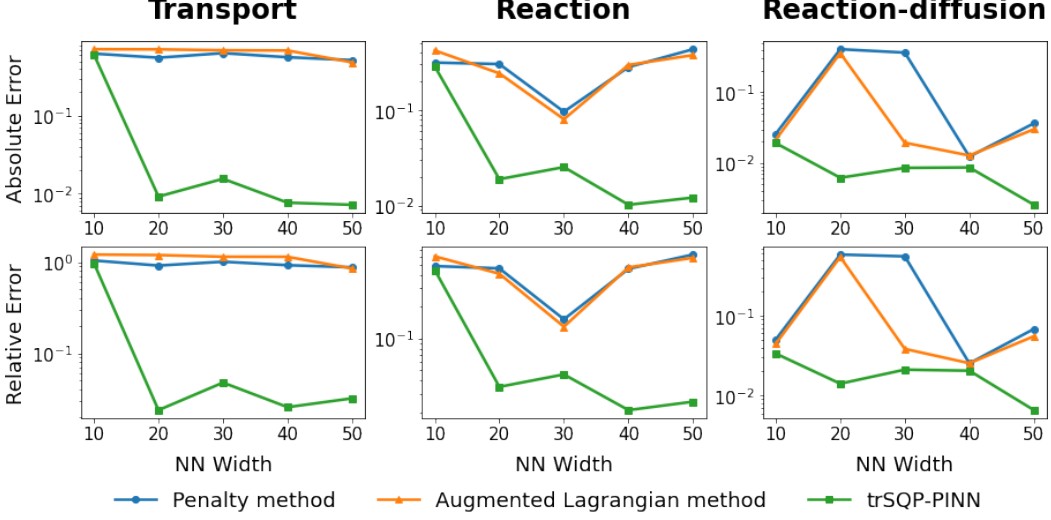

Figure 9: **Absolute and relative errors of three hard-constrained methods with varying neural networks width.** *TrSQP-PINN consistently outperforms the other methods and maintains low errors even when trained on narrower neural networks. All methods exhibit high errors when learning transport and reaction equations with only 10 neurons per layer, suggesting that the network may lack sufficient expressive power to capture the PDE solution in this case. When learning the reaction-diffusion equation, the penalty and augmented Lagrangian methods perform comparably to trSQP-PINN at neural network widths of 10 or 40. However, these methods are highly unstable, exhibiting very high errors at widths of 20 or 30, whereas trSQP-PINN maintains low errors across all widths.*

