# OpenReview forum: "Physics-Informed Neural Networks with Trust-Region Sequential Quadratic Programming"
_ICLR.cc/2025/Conference — ICLR 2025 Conference Withdrawn Submission_

### Official Review · Reviewer_ppgL · 2024-10-29

**Soundness:** 2
**Presentation:** 4
**Contribution:** 2
**Rating:** 3
**Confidence:** 3

**Summary:**

The proposed paper study the use of SQP method in the context of PINN. In a first section, the authors introduce the algorithm and detail the choices they’ve made to reach a functional algorithm for PINNs. Finally, they show the performances of their algorithm on several benchmarks.

**Strengths:**

- The paper is well written despite the numerous steps of the algorithm: several choices, approximation or constraints have been considered in the paper.
- The algorithm is well detailed and justified.
- Moreover, in the experiments, several ablations are conducted to show the robustness of the proposed method.

**Weaknesses:**

-	The proposed contributions are very specific to the PINNs case. In my opinion, it is difficult to understand in which context this method could be used in practice.
-	Misunderstanding of PINNs: PINNs designed commonly the work proposed by Raissi [1] which is completely data-free. In the paper, the author stated that PINNs are data driven which might be confusing for the reader, whether they designate [1] or Physics-informed method in general. Physics-Informed Neural Network do not rely on any data and optimize the solution neural network through the residual of the PDE and the boundary loss only (eq. 3 only in the paper).
-	Missing baselines: Since this work targets PINNs, I think more baselines from this literature should be added, first one being [1]. Other example could include soft-constrained methods, in order to highlight the interest of hard-constrained method w.r.t soft-constrained. This is unclear to me why one should take this entire algorithm if easier to implement method could be used. Moreover, since in this method data are used, data-based method should be used as a comparison or at least hybrid method. As an example, the method of [3] could be used as a base model to evaluate the performances of the proposed method, since the section 4.1 take this work as a starting point.
-	Missing experiments: The author claimed an improved training time w.r.t other method, however no computational time comparison nor number of steps to convergence is shown in the experiments.
-	Pretraining: The pretraining of this method is directly a PINNs optimization with L-BFGS, which has been proposed in [2].

1] M. Raissi, P. Perdikaris, and G.E. Karniadakis. Physics-informed neural networks: A deep learning
framework for solving forward and inverse problems involving nonlinear partial differential
equations. Journal of Computational Physics, 378:686–707, 2019. ISSN 0021-9991. doi: https://
doi.org/10.1016/j.jcp.2018.10.045

[2] Pratik Rathore, Weimu Lei, Zachary Frangella, Lu Lu, and Madeleine Udell. Challenges in
training pinns: A loss landscape perspective. In Forty-first International Conference on Ma-
chine Learning, ICML 2024

[3] Aditi Krishnapriyan, Amir Gholami, Shandian Zhe, Robert Kirby, and Michael W Mahoney. Char-
acterizing possible failure modes in physics-informed neural networks. Advances in Neural
Information Processing Systems, 34:26548–26560, 2021.

**Questions:**

-	Experiments:
   o	How long have you trained your method? What are the inference time?
   o	How doest this method behaves/scales on 2d problems?
   o	Could the author explain with more details Figure 1? More specifically, why the method performs sometimes better on negative parameters compared to positives? Intuitively, with equivalent magnitude, one would expect the performances to be as bad in both directions (if the effect of the coefficient is indeed symmetric, however, no comment is made about this aspect in the experiments).
-	Why one should use hard constrained method in the PINNs framework? I’ve noted the toy example in the footnote which is well-informative, however is this case being reached with PINNs?
-	Why does only computing an approximation of the problem is sufficient? Are the theoretical guarantees kept?

Other remarks
-	Preconditionning: PINNs are known to be ill-conditioned even with small $\mu$ coefficients, ie the differential operator in itself is ill-conditioned. This has been shown in several works such as [2, 4] and many others. Weight schedules are proposed for example in [5].

[4] Tim De Ryck, Florent Bonnet, Siddhartha Mishra, and Emmanuel de Bézenac. An operator precondi-
tioning perspective on training in physics-informed machine learning, 2023.

[5] Sifan Wang, Xinling Yu, and Paris Perdikaris. When and why pinns fail to train: A neural tangent
kernel perspective. Journal of Computational Physics

---

### Official Review · Reviewer_39eL · 2024-10-31

**Soundness:** 2
**Presentation:** 3
**Contribution:** 2
**Rating:** 3
**Confidence:** 4

**Summary:**

This paper proposed a novel training method for PINNs that aims to address the well-known failure modes of PINNs. They utilized the trust-region sequential quadratic programming to approximate the hard constraints in PINN formulation. The proposed method is assessed through numerical examples compared with the penalty method and the augmented Lagrangian method.

**Strengths:**

- Successfully implemented hard constraints within PINNs.
- Mitigated the ill-conditioning issue by alleviating the penalty parameter.
- Numerical experiments show that the proposed method outperforms traditional approaches.

**Weaknesses:**

- The problem formulation for PINNs appears unconventional.
- The experiments are insufficient to fully support the authors' claims.
- The paper lacks substantial theoretical contributions.
- There are errors and omissions in the references.

**Questions:**

1. Problem formulation
This paper considered the forward problem of partial differential equations (PDEs), which is well-posed when given appropriate initial and boundary conditions. However, the authors have formulated the problem as:
\begin{equation}\min_{\theta} \mathcal{l}(\theta) + \mu \Vert c(\theta)\Vert^2, \end{equation}
where $\mathcal{l}$ represents an "empirical" loss term, used to denote additional data, and $c(\theta)$ includes the governing equations, initial conditions, and boundary conditions. Typically, the loss function $\mathcal{l}(\theta)$ is used in inverse problems, where information is missing within the PDE framework. In the most relevant reference, [1], $\mathcal{l}(\theta)$ serves as the loss function specifically for initial and boundary conditions. Consequently, I believe that in this paper's forward problem context, initial and boundary conditions would be more appropriate choices for the hard constraints rather than $\mathcal{l}(\theta)$.

2. Experiments
The authors provide a detailed examination of PINNs’ failure modes, drawing inspiration from prior works such as [1], particularly regarding the transport equation. However, the experiments are limited to comparisons with the penalty method and the augmented Lagrangian method. To better assess the proposed method, a broader comparison with recently proposed methods, including [1,2], would be valuable. Additionally, the experiments are confined to simple, one-dimensional problems. Extending the numerical validation to more complex cases would further support the robustness of the findings.

3. Theoretical contributions
Both baseline algorithms—the penalty and augmented Lagrangian methods—come with convergence guarantees. For example, see [3] for convergence proof on the augmented Lagrangian method. Could similar theoretical contributions be established for the proposed method?

4. References
Key baseline algorithms are not cited in the manuscript (such as [3, 4] and several others proposing the augmented Lagrangian method for PINNs). Moreover, [5] did not actually consider PINNs, contrary to what the authors mentioned. It would be beneficial for the authors to carefully review their references.

[1] A. S. Krishnapriyan, A. Gholami, S. Zhe, R. M. Kirby, and M. W. Mahoney, Characterizing possible failure modes in physics-informed neural networks. NeurIPS. 2021.
[2] S. Wang, S. Sankaran, and P. Perdikaris, Respecting causality for training physics-informed neural networks. Computer Methods in Applied Mechanics and Engineering. 2024.
[3] H. Son, S. W. Cho, and H. J. Hwang, Enhanced physics-informed neural networks with augmented Lagrangian relaxation method (AL-PINNs). Neurocomputing. 2023.
[4] J. Huang, H. Wang, and Tao Zhou, An Augmented Lagrangian Deep Learning Method for Variational Problems with Essential Boundary Conditions. Communications in Computational Physics. 2022.
[5] Y. Nandwani, A. Pathak, Mausam and P. Singla, A Primal-Dual Formulation for Deep Learning with Constraints. NeurIPS. 2019.

---

### Official Review · Reviewer_k6bH · 2024-11-03

**Soundness:** 2
**Presentation:** 2
**Contribution:** 2
**Rating:** 3
**Confidence:** 3

**Summary:**

This paper focuses on the optimization problems in PINN methods. The authors introduce the trust-region sequential quadratic programming to solve the constraint in PINN, e.g., initial condition and PDE constraint. They describe the method with mathematical formulas and pseudo codes. The authors conduct experiments on the well-known failure cases of vanilla PINN. The results show that the proposed method can solve these failure cases of PINN.

**Strengths:**

1. The idea of using constraint optimization to solve PINN is interesting.
2. The authors gives a detailed description of their method, making it easy to follow.

**Weaknesses:**

1. The primary concern from the reviewer is the problem setting. According to Eq.(2), (3), the authors regard the empirical loss as loss function and regard initial/boundary condition as the constraint, so they includes several labeled observations in the training to compute the loss function. However, this setting is significantly different from the failure cases in Ref[1], in which the boundary condition and initial condition are the only available observations. Given that the observations in the vanilla PINN are viewed in the same way, i.e., using L2 loss to minimize the difference, it's hard to believe that one can observe failure cases in Ref[1] within this setting. The reviewer wonders why the authors can observe the same failure case as in Ref[1]. Also, the reviewer wonders if these failure cases can be solved by simply reweighting the loss term. If so, the experiments in current manuscript may not be suitable to demonstrate the effectiveness of the proposed method.
2. Another concern from the reviewer is the performance of the baseline penalty method. According to the footnote from the authors, the only difference  between penalty method and vanilla PINN is the magnitude of penalty. In this paper, the authors also introduce a 'pretraining method',  which tries to minimize the L2 loss of constraint term. This pretraining method, which can also be viewed as a standard PINN training without bulk observations, is shown effective even for the penalty method. However, according to the footnote from the authors, this pretraining method would be very similar to the penalty method and should not lead to significant performance difference. Why does the author can observe a large difference with/without the pretraining method in the penalty method?
3. The final concern from the reviewer is about the significance of the proposed method. The authors proposed a new optimization method to deal with the constraint in PINN. However, in Ref[2], the results from an Adam+L-BFGS optimizer is comparable with the results listed in this paper. Considering about the large number of hyperparameters introduced by the proposed method, simply using adam+L-BFGS may be a better choice. The reviewer suggest that the authors should give more solid evidence that the proposed method outperforms original methods, e.g., Adam+L-BFGS.

[1] Aditi Krishnapriyan, Amir Gholami, Shandian Zhe, Robert Kirby, and Michael W Mahoney. Characterizing possible failure modes in physics-informed neural networks. Advances in Neural Information Processing Systems, 34:26548–26560, 2021.

[2] Pratik Rathore, Weimu Lei, Zachary Frangella, Lu Lu, and Madeleine Udell. Challenges in training pinns: A loss landscape perspective. arXiv preprint arXiv:2402.01868, abs/2402.01868, 2024.

**Questions:**

Listed in weakness

---

### Note · Authors · 2024-11-12

I have read and agree with the venue's withdrawal policy on behalf of myself and my co-authors.